# Endogenous DNase Activity in an Animal Model of Acute Liver Failure

**DOI:** 10.3390/ijms24032984

**Published:** 2023-02-03

**Authors:** Ľubica Janovičová, Katarína Kmeťová, Nikola Pribulová, Jakub Janko, Barbora Gromová, Roman Gardlík, Peter Celec

**Affiliations:** 1Institute of Molecular Biomedicine, Faculty of Medicine, Comenius University, 81108 Bratislava, Slovakia; 2Institute of Pathophysiology, Faculty of Medicine, Comenius University, 81108 Bratislava, Slovakia

**Keywords:** cell-free DNA, nuclease, DNA cleavage, liver damage, drug induced liver injury

## Abstract

Deoxyribonucleases (DNases) cleave extracellular DNA (ecDNA) and are under intense research as interventions for diseases associated with high ecDNA, such as acute live injury. DNase I treatment decreases morbidity and mortality in this animal model. Endogenous DNase activity has high interindividual variability. In this study, we tested the hypothesis that high endogenous DNase activity is beneficial in an animal model of acute liver failure. DNase activity was measured in the plasma of adult male mice taken before i.p. injection of thioacetamide to induce acute liver failure. The survival of mice was monitored for 48 h. Mice were retrospectively divided into two groups based on the median DNase activity assessed using the gel-based single-radial enzyme diffusion assay. In acute liver failure, mice with a higher baseline DNase activity had lower mortality after 48 h (by 25%). Different protection of ecDNA against nucleases by vesicles or DNA-binding proteins could play a role and should be further evaluated. Similarly, the role of endogenous DNase activity should be analyzed in other disease models associated with high ecDNA.

## 1. Introduction

Extracellular DNA (ecDNA) is a collective term for all DNA that can be found in the extracellular space. It can be released during apoptosis or necrosis and secreted outside of cells. It can be genomic or mitochondrial, which makes it of either nuclear or mitochondrial origin. Despite all mechanisms that ensure its removal, ecDNA can trigger an immune system response process, which is called sterile inflammation [1]. One of the mechanisms that facilitate the removal of ecDNA are deoxyribonucleases (DNases) [2]. Previously, it was observed that ecDNA is released in acute liver failure. It is suspected that removal mechanisms may not be sufficient to cleave all ecDNA that is released. The administration of DNase I was shown to improve disease markers in acute liver failure [3].

DNA is found inside living cells in the nucleus and mitochondria. When cells die, their DNA can be released into the extracellular space. This DNA is also called extracellular DNA (ecDNA). DNA outside of cells is recognized by the immune system and can cause inflammation [1]. Secreted deoxyribonucleases (DNases) cleave ecDNA rapidly [2]. However, this cleavage might be insufficient if tissues get damaged and a large amount of DNA is released. High ecDNA concentrations were described in acute liver failure, and a study has shown that the administration of DNase has a protective effect. In liver failure, ecDNA originates from dying hepatocytes [4]. The ecDNA can have different properties, such as fragment length distribution, protection against cleavage by DNases, and subcellular origin. The ecDNA is typically short, with around 165 base pairs [5]. Longer fragments of ecDNA could suggest that ecDNA is either inefficiently cleaved or protected by proteins or vesicles. The subcellular origin of ecDNA, nuclear or mitochondrial, implies the inflammatory properties and protection against DNases. Nuclear DNA can be bound to histones, which have proinflammatory properties [6]. Mitochondrial DNA is likely unprotected from being cleaved by DNases. Mitochondrial DNA can resemble bacteria in its sequence and unmethylated CpGs, and have proinflammatory properties [7]. Both nuclear and mitochondrial ecDNA can be found in microparticles [8,9], however, the inability of the blood to clear ecDNA could be caused by the characteristics of ecDNA. It was observed that there is a high interindividual variability in ecDNA concentrations in experimental animals and in humans [10,11]. DNases are important when it comes to the cleavage of ecDNA in the blood. Deficiency of DNase enzymes leads to a decreased survival of mice when coupled with inflammation and neutrophil activation [12]. Deletion of DNases may lead to a different fragment length profile and longer ecDNA fragments found in knock-out mice [13]. Additionally, high interindividual variability of DNase activity was found in experimental animals [11], however, it is unknown whether variability of endogenous DNase activity is associated with acute liver failure.

We hypothesize that diseases such as acute liver failure, where ecDNA is high, may have lower mortality rates when endogenous DNase activity in high. We propose that the removal of ecDNA from the circulation by endogenous DNases presents an advantage in acute liver failure. Besides, the state of ecDNA which can be either unprotected or protected against cleavage by DNases can differ in disease. The aim of this study was to assess the possible advantages of having high DNase activity in the plasma of a model of acute liver failure.

## 2. Results

Determination of ecDNA concentration in acute liver failure model served to support the hypothesis that DNase activity in the plasma could affect survival. The concentration of total ecDNA is higher in acute liver failure (Figure 1A). The mean concentration of total ecDNA in the control group was 22.0 ng/mL and 260.8 ng/mL in the acute liver failure group. The ecDNA variability was 120% in acute liver failure and 85% in the control group.

Median DNase activity in mouse plasma was 3.7 K.u./mL. This was used to divide mice to high DNase and low DNase activity groups. Overall mortality of mice with acute liver failure was 75%. However, only approximately 20% of mice with lower DNase activity survived acute liver failure. In comparison, approximately 50% of mice in the group had high DNase activity. Statistically, more mice have survived acute liver failure when they had high endogenous DNase activity in comparison to littermates with low DNase activity (Figure 1B; Log-rank test; *p* = 0.02).

Here the ability of ecDNA to be cleaved was assessed by the addition of DNase I into plasma samples. There was no difference between groups in the percentage of DNase-protected ecDNA after the first centrifugation (Figure 2A; *t*-test; *p* = n.s.). There was 24% ecDNA protected from being cleaved in control groups, and 29% in acute liver failure. The second centrifugation, removing larger vesicles from plasma, also had no difference between groups in the percentage of protected ecDNA (Figure 2B; *t*-test; *p* = n.s.). After the second centrifugation, the mean percentage of protected ecDNA in the plasma of control mice was 35% and in acute liver failure 16%. The difference was observed in protected ecDNA in the microparticles of mice with acute liver failure having less protected ecDNA in comparison to controls (Figure 2C; *t*-test; *p* = 0.008). The mean percentage of protected ecDNA in the microparticles was in 65% controls and 19% in acute liver failure.

The proportion of ncDNA protected from being cleaved by DNase I treatment was determined. No differences in ncDNA concentration were observed in plasma centrifuged once, with the mean percentage of protected ncDNA being 17% in controls and 17% in acute liver failure (Figure 2D; *t*-test; *p* = n.s.). High variability in ncDNA concentration was observed in plasma centrifuged twice, with the mean percentage of protected ncDNA in controls being over 100% and acute liver failure over 100% (Figure 2E; *t*-test; *p* = n.s.). Differences were observed in the concentration of protected ncDNA in microparticles, whereby it was 38% for control mice and over 100% for acute liver failure (Figure 2F; *t*-test; *p* = n.s.).

Variability in protected ecDNA was especially high in the mtDNA fraction. In many cases, the percentage of protected mtDNA was higher than 100%. The percentage of protected mtDNA in plasma centrifuged once was 36% in the control group. The percentage of protected mtDNA in acute liver failure was 36% (Figure 2G; *t*-test; *p* = n.s.). Plasma centrifuged twice contained 39% of protected mtDNA in the control group and more than 100% in acute liver failure (Figure 2H; *t*-test; *p* = n.s.). On the other hand, the percentage of protected mtDNA in the microparticles was more than 100% in the control group, but 88% in acute liver failure (Figure 2I; *t*-test; *p* = n.s.). Fragmentation analysis of the selected samples showed a fragmentation profile in the control sample with shorter fragments. However, a sample of ecDNA from a mouse with acute liver failure contained longer fragments (Figure 3).

## 3. Discussion

Data in this study describe the effects of endogenous DNase activity in an animal model of acute liver failure. High quantities of ecDNA are released in acute liver failure, and this DNA originates from dying hepatocytes [14]. The properties of this ecDNA can be different, which may prevent it to be cleaved by DNases. Those differences can be observed in sequence, protein-binding, fragmentation, origin, and immunogenicity. If ecDNA is bound to histones [15] or associated with vesicles, it may not be degraded by DNases [16]. Here we show that high endogenous DNase activity has a protective effect in acute liver failure.

The subcellular origin, nuclear or mitochondrial, can affect immunogenicity. Mitochondrial DNA is not packaged in nucleosomes, but it can associate with proteins and is more immunogenic [17]. Mitochondrial DNA has unmethylated CpG islands which are found in some bacteria and are recognized by the immune system [18]. On the other hand, nuclear DNA is tightly packed in structures that can be inaccessible to DNases. 

EcDNA can be cleaved by DNases in the blood, filtered out by kidneys, or removed by macrophages [19]. Here, only the advantages of higher DNase activity, and therefore more capacity to remove ecDNA, were assessed. It is difficult to draw a comparison between endogenous DNase activity and treatment with DNase I used in other studies. The concentration of injected DNase I is high and it can be as much as 10 mg/kg [20] or 20 mg/kg [21]. Such high concentrations may not be observed in endogenous DNase activity. High variability in endogenous DNase activity may affect the development of the disease. It is crucial to study not only ecDNA as a marker, but examine the role of DNase activity and properties of ecDNA such as its fragmentation, length and resistance to being cleaved by DNases. Description of ecDNA properties, origin, and removal, could improve its applicability as a therapeutic target in diseases associated with high concentrations of ecDNA.

## 4. Materials and Methods

### 4.1. Animals

Adult male CD-1 mice (n = 100) included in experiments were purchased from Anlab (Prague, Czech Republic). Stabile temperature and a 12-h light/ dark cycle were employed to mouse housing. They had ad libitum access to food and drinking water. All described procedures were conducted in accordance with Slovak legislation. Mice were monitored throughout experiments and had their well-being score assessed frequently to meet the humane endpoints. The animals were housed in compliance with the ethics committee of the Institute of Molecular Biomedicine, Comenius University in Bratislava (Ro-536/18-221/3).

Baseline blood was collected for endogenous DNase activity measurement into heparin-containing tubes. Fifty mice had induced acute liver failure by a single i.p. injection of thioacetamide (300 mg/kg). All mice were monitored for 48 h. Mice that survived the 48 h were sacrificed using a cervical dislocation procedure. Baseline blood was centrifuged at 1600× *g* for 10 min at 4 °C. Obtained plasma was stored until subsequent analyses at −20 °C. Based on the DNase activity measurement, mice were retrospectively divided into two groups: those that had high DNase activity and those that had low DNase activity. 

In experiments where the proportion of protected and unprotected DNase was estimated, 15 adult CD-1 male mice were included. Mice that had induced liver failure were i.p. injected with thioacetamide (300 mg/kg). The control mice were i.p. injected with saline. Mice were sacrificed 24 h after induction of acute liver failure. The blood of these mice was collected in heparin-containing tubes and centrifuged at 1600× *g* for 10 min at 4 °C. Samples were split before the second centrifugation and one half was centrifuged again at 16,000× *g* for 10 min at 4 °C. Plasma samples, both which were centrifuged the second time and those which were not, were split again and one half was incubated with DNase I (20 K.u. per sample) for 30 min at 37 °C. The reaction was stopped by the addition of protease and subsequent isolation of ecDNA. The ecDNA was isolated using the QIAamp DNA Blood Mini kit with standard protocol. The concentration of ecDNA was quantified using Qubit fluorometer and Qubit DNA High Sensitivity kit. Analysis of the distribution of ecDNA fragment length of the selected samples from males and castrated males was done using capillary electrophoresis and the BIABooster technology (Adelis, Grabels, France) [22].

### 4.2. Real-Time PCR

EcDNA of two subcellular origins was quantified using nuclear-specific and mitochondrial-specific primers. The master mix qPCR SybrMaster (Jena Bioscience) was used to quantify ecDNA. The real-time PCR started with initial melting at 95 °C and was then followed by 40 amplification cycles. The amplification cycle consisted of melting at 95 °C for 15 s and annealing with extension at 60 °C for 1 min. The final concentration of primers in a reaction was 250 pg. The used primers for the quantification of ncDNA were forward: 5′-TGTCAGATATGTCCTTCAGCAAGG-3′ and reverse: 5′-TGCTTAACTCTGCAGGCGTATG-3′. The mtDNA was quantified using forward primer: 5′-CCCAGCTACTACCATCATTCAAGT-3′ and reverse: 5′-GATGGTTTGGGAGATTGGTTGATGT-3′. 

### 4.3. DNase Activity

DNase activity was measured using a modified single-radial enzyme-diffusion assay. Shortly, gels contained 2 mM CaCl_2_, 2 mM MgCl_2_, 20 mM Tris HCl, 0.5 mg/mL of isolated DNA from chicken liver and Goodview fluorescent dye in the final concentration. Gels were incubated for 16 h at 37 °C. Pictures were made using UVP BioDoc-It imaging system, and analyzed using the ImageJ software.

### 4.4. Statistics

Groups were compared using the one-way ANOVA test (α = 0.05), with Bonferroni correction. Survivals are shown on Kaplan-Meier plots, and the survival of high DNase and low DNase groups were compared using the log-rank test (α = 0.05). The comparisons are annotated with statistics where α = 0.05 is *, α = 0.01 is ** and non-significant results are annotated as n.s. or ns. Analysis was performed using the GraphPad Prism 6.

## Figures and Tables

**Figure 1 ijms-24-02984-f001:**
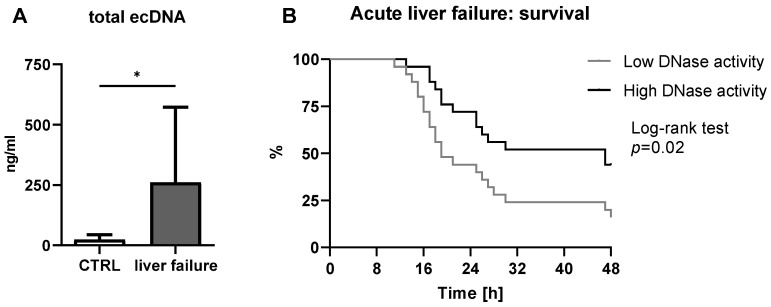
(**A**) The extracellular DNA (ecDNA) is higher in acute liver failure in comparison to the control group (unpaired *t*-test; *p* = 0.01; t = 2.74). (**B**) The survival of mice with low and high endogenous DNase activity were assessed (log-rank test; *p* = 0.02). *—*t*-test with *p* < 0.05.

**Figure 2 ijms-24-02984-f002:**
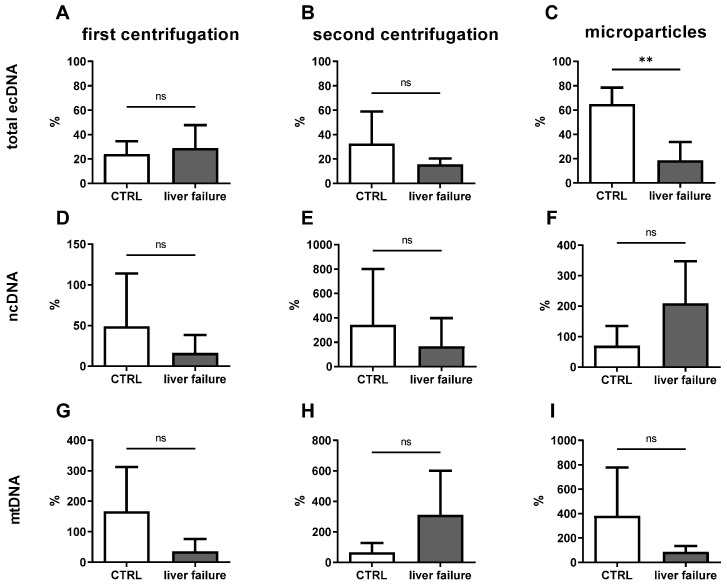
The proportion of protected ecDNA in acute liver failure in comparison to the control group in the fraction after centrifugation at (**A**) 1600× *g*, (**B**) 16,000× *g*, and (**C**) in the microparticles. The percentage of protected ncDNA was determined for all fractions, including plasma centrifuged at (**D**) 1600× *g*, (**E**) 16,000× *g*, and (**F**) the microparticles. The same was measured for the mtDNA from plasma centrifuged at (**G**) 1600× *g*, (**H**) 16,000× *g*, and (**I**) the microparticles. Data are presented as a mean with standard deviation. **—*t*-test with *p* < 0.01, ns—non-significant.

**Figure 3 ijms-24-02984-f003:**
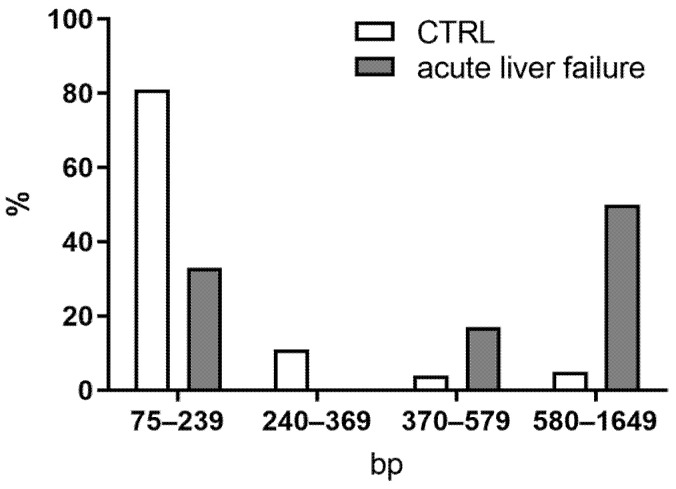
To assess the fragmentation profile of ecDNA, fragment lengths were analyzed to compare control and acute liver failure groups.

## Data Availability

Data are available from the authors upon reasonable request.

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
