# Peer review of "Endogenous DNase Activity in an Animal Model of Acute Liver Failure"

_ijms, 2023, doi:10.3390/ijms24032984_

Round 1

Reviewer 1 Report

The authors tested hypothesis whether high endogenous deoxyribonucleases (DNase) activity is beneficial in animal model of acute liver failure. Liver failure in mice was induced using thioacetamide. They observed that mice with higher concentrations of the DNase activity had lower mortality after 48 hours than the control group. The manuscript is clearly written, although some more background could be presented that is directly linked to the analysis of the results. Below are some minor remarks regarding the manuscript.

Explain why the proportions of protected ecDNA are higher than 100% on some of the plots (Figure 2).

Provide explanation why more microparticles are expected/observed in the control group than in the “liver failure” group.

Write in the figure captions what the whiskers in the plots represent.

What do top-lines (ns) in figure 2 indicate, (*) in Figure 1?

Correct …of select samples… > > …of the selected samples… (line 170)

Rewrite the sentence “The reaction contained 0.25 ul of primers in a concentration 10 nM/ml.” (line 178)

t-test results: what does p=n.s. mean.?

Increase font size in Figures 1 and 2.

Author Response

Dear editorial board,

We would like to thank both reviewers for their suggestions and comments. We tried to apply all suggestions and solve all raised issues. Here is our point by point response.

Reviewer 1:

The authors tested hypothesis whether high endogenous deoxyribonucleases (DNase) activity is beneficial in animal model of acute liver failure. Liver failure in mice was induced using thioacetamide. They observed that mice with higher concentrations of the DNase activity had lower mortality after 48 hours than the control group. The manuscript is clearly written, although some more background could be presented that is directly linked to the analysis of the results. Below are some minor remarks regarding the manuscript.

Our response:

We would like to thank reviewer 1 for all his/her feedback on our manuscript. We believe that the research concerning ecDNA and DNase activity in relation to animal models of diseases is important. It is not only a search for the a better or more sensitive biomarkers, but also about the understanding of the disease mechanisms.

Explain why the proportions of protected ecDNA are higher than 100% on some of the plots (Figure 2).

Our response:

We thank the reviewer for the question and interest in our results. It is expected that DNase I treatment cleaves ecDNA in plasma and so, the concentration should decrease. However, in some cases, the isolation of ecDNA is more successful with higher yields after adding DNase I. We think that this is a technical issue rather than a biological phenomenon. We also lack clear data supporting this claim. That is why we did not go into detail of this outcome. We surely will look into it because we would like to study ecDNA associated with exosomes where this seems to be a rule rather than an exception, but within this experiment it is difficult to cover this pure speculation.

Provide explanation why more microparticles are expected/observed in the control group than in the “liver failure” group.

Our response:

Our short and honest answer is that we do not know. We can only speculate and have no data to support this. However, we think that it might be related to the source of ecDNA in liver failure vs controls, as well as to the the action of endogenous DNases that might be different before and after disease model induction. Also, the protection mechanisms – vesicles vs DNA-binding proteins might differ between the groups. However, none of this is tested and rather than add more bias to the literature we would prefer to gather more reasonable data and prepare another manuscript with tested hypotheses.

Write in the figure captions what the whiskers in the plots represent.

Our response:

We thank the reviewer for the comment. We fixed the missing captions in the manuscript.

What do top-lines (ns) in figure 2 indicate, (*) in Figure 1?

Our response:

We added the missing annotations to the figures and the statistics section.

Correct …of select samples… > > …of the selected samples… (line 170)

Our response:

We corrected this in the updated version of the manuscript.

Rewrite the sentence “The reaction contained 0.25 ul of primers in a concentration 10 nM/ml.”(line 178)

Our response:

As suggested we have rewritten the sentence.

t-test results: what does p=n.s. mean.?

Our response:

We have added the explanation for the “n.s.” in the manuscript as this stands for non-significant.

Increase font size in Figures 1 and 2.

Our response:

We thank the reviewer for the suggestion to increase the font sizes in our figures. We fixed figures in the updated version of the manuscript.

Reviewer 2 Report

In the manuscript entitled “Endogenous DNase activity in animal model of acute liver failure” the authors describe the role of ecDNA in acute liver failure and a probable role of DNAse in suppressing ecDNA mediated inflammation is discussed. According to my analysis, these findings are extremely limited showing a bare correlation between increased DNAse activity and less mortality in TAA model. Higher ECDNA is associated with less DNAse activity. But mechanistic insight is lacking. Most of the data is simply correlative.

The authors claim that this is the first report of the association of DNase activity and survival in an animal model of acute liver failure. It would have been better if a more detailed analysis of this correlation could be done. As of now, the findings are of limited relevance.

Author Response

Dear editorial board,

We would like to thank both reviewers for their suggestions and comments. We tried to apply all suggestions and solve all raised issues. Here is our point by point response.

Reviewer 2:

In the manuscript entitled “Endogenous DNase activity in animal model of acute liver failure” the authors describe the role of ecDNA in acute liver failure and a probable role of DNAse in suppressing ecDNA mediated inflammation is discussed. According to my analysis, these findings are extremely limited showing a bare correlation between increased DNAse activity and less mortality in TAA model. Higher ECDNA is associated with less DNAse activity. But mechanistic insight is lacking. Most of the data is simply correlative. The authors claim that this is the first report of the association of DNase activity and survival in an animal model of acute liver failure. It would have been better if a more detailed analysis of this correlation could be done. As of now, the findings are of limited relevance.

Our response:

We would like to thank reviewer 2 for reviewing our manuscript and for his/her critical evaluation. We agree and admit that the data are mostly correlative and do not provide mechanistic details. On the other hand, which mechanisms are the ones that are worthy studying depends more or less only on associations reported in the literature. If these are not reported than it gets difficult. A human genetic model of DNase deficiency is rare and is potentially affected by adaptations of other nucleases and their activity. We have previously described a huge biological variability in total DNase activity even in healthy subjects. We still do not know the sources – reasons for this variability, but they seem to be epigenetic in the wider meaning of the word. The fact that the found endogenous variability of DNase activity has consequences in an induced disease model shows that the variability is worthy further investigations by us, but also by others. We think that it might be of importance to study the same phenomenon in other disease models – sepsis, hemolytic anemia, ischemia-reperfusion injury etc. However, the search for the sources of variability seems to be also of relevance given the outcomes of the presented experiment. We, however, totally agree with the reviewer regarding the “limited relevance” of the findings. We are aware of the limitation and discuss the most important ones in the manuscript. We, nevertheless, think that all experiments have “limited relevance”, but could guide more relevant clinical studies.

Round 2

Reviewer 2 Report

None